# Offline RL Without Off-Policy Evaluation

**David Brandfonbrener**     **William F. Whitney**     **Rajesh Ranganath**     **Joan Bruna**
Department of Computer Science, Center for Data Science
New York University
david.brandfonbrener@nyu.edu

## Abstract

Most prior approaches to offline reinforcement learning (RL) have taken an iterative actor-critic approach involving off-policy evaluation. In this paper we show that simply doing one step of constrained/regularized policy improvement using an on-policy Q estimate of the behavior policy performs surprisingly well. This one-step algorithm beats the previously reported results of iterative algorithms on a large portion of the D4RL benchmark. The one-step baseline achieves this strong performance while being notably simpler and more robust to hyperparameters than previously proposed iterative algorithms. We argue that the relatively poor performance of iterative approaches is a result of the high variance inherent in doing off-policy evaluation and magnified by the repeated optimization of policies against those estimates. In addition, we hypothesize that the strong performance of the one-step algorithm is due to a combination of favorable structure in the environment and behavior policy.

## 1 Introduction

An important step towards effective real-world RL is to improve sample efficiency. One avenue towards this goal is offline RL (also known as batch RL) where we attempt to learn a new policy from data collected by some other behavior policy without interacting with the environment. Recent work in offline RL is well summarized by Levine et al. [2020].

In this paper, we challenge the dominant paradigm in the deep offline RL literature that primarily relies on actor-critic style algorithms that alternate between policy evaluation and policy improvement [Fujimoto et al., 2018a, 2019, Peng et al., 2019, Kumar et al., 2019, 2020, Wang et al., 2020b, Wu et al., 2019, Kostrikov et al., 2021, Jaques et al., 2019, Siegel et al., 2020, Nachum et al., 2019]. All these algorithms rely heavily on off-policy evaluation to learn the critic. Instead, we find that a simple baseline which only performs one step of policy improvement using the behavior Q function often outperforms the more complicated iterative algorithms. Explicitly, we find that our one-step algorithm beats prior results of iterative algorithms on most of the gym-mujoco [Brockman et al., 2016] and Adroit [Rajeswaran et al., 2017] tasks in the the D4RL benchmark suite [Fu et al., 2020].

We then dive deeper to understand why such a simple baseline is effective. First, we examine what goes wrong for the iterative algorithms. When these algorithms struggle, it is often due to poor off-policy evaluation leading to inaccurate Q values. We attribute this to two causes: (1) distribution shift between the behavior policy and the policy to be evaluated, and (2) iterative error exploitation whereby policy optimization introduces bias and dynamic programming propagates this bias across the state space. We show that empirically both issues exist in the benchmark tasks and that one way to avoid these issues is to simply avoid off-policy evaluation entirely.

Finally, we recognize that while the the one-step algorithm is a strong baseline, it is not always the best choice. In the final section we provide some guidance about when iterative algorithms can perform better than the simple one-step baseline. Namely, when the dataset is large and behavior policy has good coverage of the state-action space, then off-policy evaluation can succeed and iterative

35th Conference on Neural Information Processing Systems (NeurIPS 2021).

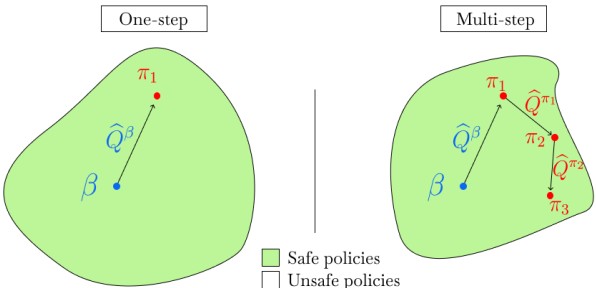

Figure 1: A cartoon illustration of the difference between one-step and multi-step methods. All algorithms constrain themselves to a neighborhood of "safe" policies around $\beta$. A one-step approach (left) only uses the on-policy $\widehat{Q}^\beta$, while a multi-step approach (right) repeatedly uses off-policy $\widehat{Q}^{\pi_i}$.

algorithms can be effective. In contrast, if the behavior policy is already fairly good, but as a result does not have full coverage, then one-step algorithms are often preferable.

Our main contributions are:

- A demonstration that a simple baseline of one step of policy improvement outperforms more complicated iterative algorithms on a broad set of offline RL problems.

- An examination of failure modes of off-policy evaluation in iterative offline RL algorithms.

- A description of when one-step algorithms are likely to outperform iterative approaches.

## 2   Setting and notation

We will consider an offline RL setup as follows. Let $\mathcal{M} = \{\mathcal{S}, \mathcal{A}, \rho, P, R, \gamma\}$ be a discounted infinite-horizon MDP. In this work we focus on applications in continuous control, so we will generally assume that both $\mathcal{S}$ and $\mathcal{A}$ are continuous and bounded. We consider the offline setting where rather than interacting with $\mathcal{M}$, we only have access to a dataset $D_N$ of $N$ tuples of $(s_i, a_i, r_i)$ collected by some behavior policy $\beta$ with initial state distribution $\rho$. Let $r(s, a) = \mathbb{E}_{r|s,a}[r]$ be the expected reward. Define the state-action value function for any policy $\pi$ by $Q^\pi(s, a) := \mathbb{E}_{P,\pi|s_0=s,\ a_0=a}[\sum_{t=0}^\infty \gamma^t r(s_t, a_t)]$. The objective is to maximize the expected return $J$ of the learned policy:

$$J(\pi) := \mathop{\mathbb{E}}_{\rho, P, \pi} \left[ \sum_{t=0}^\infty \gamma^t r(s_t, a_t) \right] = \mathop{\mathbb{E}}_{\substack{s \sim \rho \\ a \sim \pi|s}} [Q^\pi(s, a)]. \tag{1}$$

Following Fu et al. [2020] and others in this line of work, we allow access to the environment to tune a small ($< 10$) set of hyperparameters. See Paine et al. [2020] for a discussion of the active area of research on hyperparameter tuning for offline RL. We also discuss this further in Appendix C.

## 3   Related work

**Iterative algorithms.**   Most prior work on deep offline RL consists of iterative actor-critic algorithms. The primary innovation of each paper is to propose a different mechanism to ensure that the learned policy does not stray too far from the data generated by the behavior policy. Broadly, we group these methods into three camps: policy constraints/regularization, modifications of imitation learning, and Q regularization:

1. The majority of prior work acts directly on the policy. Some authors have proposed explicit constraints on the learned policy to only select actions where $(s, a)$ has sufficient support under the data generating distribution [Fujimoto et al., 2018a, 2019, Laroche et al., 2019]. Another proposal is to regularize the learned policy towards the behavior policy [Wu et al., 2019] usually either with a KL divergence [Jaques et al., 2019] or MMD [Kumar et al., 2019]. This is a very straighforward way to stay close to the behavior with a hyperparameter that determines just how close. All of these algorithms are iterative and rely on off-policy evaluation.

2. Siegel et al. [2020], Wang et al. [2020b], Chen et al. [2020] all use algorithms that filter out datapoints with low Q values and then perform imitation learning. Wang et al. [2018], Peng et al. [2019] use a weighted imitation learning algorithm where the weights are determined by exponentiated Q values. These algorithms are iterative.

3. Another way to prevent the learned policy from choosing unknown actions is to incorporate some form of regularization to encourage staying near the behavior and being pessimistic about unknown state, action pairs [Wu et al., 2019, Nachum et al., 2019, Kumar et al., 2020, Kostrikov et al., 2021, Gulcehre et al., 2021]. However, being able to properly quantify uncertainty about unknown states is notoriously difficult when dealing with neural network value functions [Buckman et al., 2020].

**One-step algorithms.**  Some recent work has also noted that optimizing policies based on the behavior value function can perform surprisingly well. As we do, Goo and Niekum [2020] studies the continuous control tasks from the D4RL benchmark, but they examine a complicated algorithm involving ensembles, distributional Q functions, and a novel regularization technique. In contrast, we analyze a substantially simpler algorithm and get better performance on the D4RL tasks. We also focus more of our contribution on understanding and explaining this performance. Gulcehre et al. [2021] studies the discrete action setting and finds that a one-step algorithm (which they call "behavior value estimation") outperforms prior work on Atari games and other discrete action tasks from the RL Unplugged benchmark [Gulcehre et al., 2020]. They also introduce a novel regularizer for the evaluation step. In contrast, we consider the continuous control setting. This is a substantial difference in setting since continuous control requires actor-critic algorithms with parametric policies while in the discrete setting the policy improvement step can be computed exactly from the Q function. Moreover, while Gulcehre et al. [2021] attribute the poor performance of iterative algorithms to "overestimation", we define and separate the issues of distribution shift and iterative error exploitation which can combine to cause overestimation. This separation helps to expose the difference between the fundamental limits of off-policy evaluation from the specific problems induced by iterative algorithms, and will hopefully be a useful distinction to inspire future work. Finally, a one-step variant is also briefly discussed in Nadjahi et al. [2019], but is not the focus of that work.

There are also important connections between the one-step algorithm and the literature on conservative policy improvement [Kakade and Langford, 2002, Schulman et al., 2015, Achiam et al., 2017], which we discuss in more detail in Appendix B.

## 4   Defining the algorithms

In this section we provide a unified algorithmic template for model-free offline RL algorithms as offline approximate modified policy iteration. We show how this template captures our one-step algorithm as well as a multi-step policy iteration algorithm and an iterative actor-critic algorithm. Then any choice of policy evaluation and policy improvement operators can be used to define one-step, multi-step, and iterative algorithms.

### 4.1   Algorithmic template

We consider a generic offline approximate modified policy iteration (OAMPI) scheme, shown in Algorithm 1 (and based off of Puterman and Shin [1978], Scherrer et al. [2012]). Essentially the algorithm alternates between two steps. First, there is a policy evaluation step where we estimate the Q function of the current policy $\pi_{k-1}$ by $\widehat{Q}^{\pi_{k-1}}$ using only the dataset $D_N$. Implementations also often use the prior Q

---

**Algorithm 1:** OAMPI

**input :** $K$, dataset $D_N$, estimated behavior $\hat{\beta}$
Set $\pi_0 = \hat{\beta}$. Initialize $\widehat{Q}^{\pi_{-1}}$ randomly.
**for** $k = 1, \ldots, K$ **do**
  Policy evaluation: $\widehat{Q}^{\pi_{k-1}} = \mathcal{E}(\pi_{k-1}, D_N, \widehat{Q}^{\pi_{k-2}})$
  Policy improvement: $\pi_k = \mathcal{I}(\widehat{Q}^{\pi_{k-1}}, \hat{\beta}, D_N, \pi_{k-1})$
**end**

---

estimate $\widehat{Q}^{\pi_{k-2}}$ to warm-start the approximation process. Second, there is a policy improvement step. This step takes in the estimated Q function $\widehat{Q}^{\pi_{k-1}}$, the estimated behavior $\hat{\beta}$, and the dataset $D_N$ and produces a new policy $\pi_k$. Again an algorithm may use $\pi_{k-1}$ to warm-start the optimization. Moreover, we expect this improvement step to be regularized or constrained to ensure that $\pi_k$ remains in the support of $\beta$ and $D_N$. Choices for this step are discussed below. Now we discuss a few ways to instantiate the template.

**One-step.** The simplest algorithm sets the number of iterations $K = 1$. We learn $\hat{\beta}$ by maximum likelihood and train the policy evaluation step to estimate $Q^\beta$. Then we use any one of the policy improvement operators discussed below to learn $\pi_1$. Importantly, this algorithm completely avoids off-policy evaluation.

**Multi-step.** The multi-step algorithm now sets $K > 1$. The evaluation operator must evaluate off-policy since $D_N$ is collected by $\beta$, but evaluation steps for $K \geq 2$ require evaluating policies $\pi_{k-1} \neq \beta$. Each iteration is trained to convergence in both the estimation and improvement steps.

**Iterative actor-critic.** An actor critic approach looks somewhat like the multi-step algorithm, but does not attempt to train to convergence at each iteration and uses a much larger $K$. Here each iteration consists of one gradient step to update the Q estimate and one gradient step to improve the policy. Since all of the evaluation and improvement operators that we consider are gradient-based, this algorithm can adapt the same evaluation and improvement operators used by the multi-step algorithm. Most algorithms from the literature fall into this category [Fujimoto et al., 2018a, Kumar et al., 2019, 2020, Wu et al., 2019, Wang et al., 2020b, Siegel et al., 2020].

## 4.2 Policy evaluation operator

Following prior work on continuous state and action problems, we always evaluate by simple fitted Q evaluation [Fujimoto et al., 2018a, Kumar et al., 2019, Siegel et al., 2020, Wang et al., 2020b, Paine et al., 2020, Wang et al., 2021]. In practice this is optimized by TD-style learning with the use of a target network [Mnih et al., 2015] as in DDPG [Lillicrap et al., 2015]. We do not use any double Q learning or Q ensembles [Fujimoto et al., 2018b]. For the one-step and multi-step algorithms we train the evaluation procedure to convergence on each iteration and for the iterative algorithm each iteration takes a single stochastic gradient step. See Voloshin et al. [2019], Wang et al. [2021] for more comprehensive examinations of policy evaluation and some evidence that this simple fitted Q iteration approach is reasonable. It is an interesting direction for future work to consider other operators that use things like importance weighting [Munos et al., 2016] or pessimism [Kumar et al., 2020, Buckman et al., 2020].

## 4.3 Policy improvement operators

To instantiate the template, we also need to choose a specific policy improvement operator $\mathcal{I}$. We consider the following improvement operators selected from those discussed in the related work section. Each operator has a hyperparameter controlling deviation from the behavior policy.

**Behavior cloning.** The simplest baseline worth including is to just return $\hat{\beta}$ as the new policy $\pi$. Any policy improvement operator ought to perform at least as well as this baseline.

**Constrained policy updates.** Algorithms like BCQ [Fujimoto et al., 2018a] and SPIBB [Laroche et al., 2019] constrain the policy updates to be within the support of the data/behavior. In favor of simplicity, we implement a simplified version of the BCQ algorithm that removes the "perturbation network" which we call Easy BCQ. We define a new policy $\hat{\pi}_k^M$ by drawing $M$ samples from $\hat{\beta}$ and then executing the one with the highest value according to $\widehat{Q}^\beta$. Explicitly:

$$\hat{\pi}_k^M(a|s) = \mathbb{1}[a = \arg\max_{a_j}\{\widehat{Q}^{\pi_{k-1}}(s, a_j) : a_j \sim \pi_{k-1}(\cdot|s),\ 1 \leq j \leq M\}]. \tag{2}$$

**Regularized policy updates.** Another common idea proposed in the literature is to regularize towards the behavior policy [Wu et al., 2019, Jaques et al., 2019, Kumar et al., 2019]. For a general divergence $D$ we can define an algorithm that maximizes a regularized objective:

$$\hat{\pi}_k^\alpha = \arg\max_\pi \sum_i \mathbb{E}_{a \sim \pi|s}[\widehat{Q}^{\pi_{k-1}}(s_i, a)] - \alpha D(\hat{\beta}(\cdot|s_i), \pi(\cdot|s_i)) \tag{3}$$

A comprehensive review of different variants of this method can be found in Wu et al. [2019] which does not find dramatic differences across regularization techniques. In practice, we will use reverse KL divergence, i.e. $KL(\pi(\cdot|s_i)\|\hat{\beta}(\cdot|s_i))$. To compute the reverse KL, we draw samples from $\pi(\cdot|s_i)$ and use the density estimate $\hat{\beta}$ to compute the divergence. Intuitively, this regularization forces $\pi$ to remain within the support of $\beta$ rather than incentivizing $\pi$ to cover $\beta$.

**Variants of imitation learning.** Another idea, proposed by [Wang et al., 2018, Siegel et al., 2020, Wang et al., 2020b, Chen et al., 2020] is to modify an imitation learning algorithm either by filtering or weighting the observed actions to incentivize policy improvement. The weighted version that we implement uses exponentiated advantage estimates to weight the observed actions:

$$\hat{\pi}_k^\tau = \arg\max_\pi \sum_i \exp(\tau(\widehat{Q}^{\pi_{k-1}}(s_i, a_i) - \widehat{V}(s_i))) \log \pi(a_i|s_i). \tag{4}$$

With these definitions, we can now move on to testing various combinations of algorithmic template (one-step, multi-step, or iterative) and improvement operator (Easy BCQ, reverse KL regularization, or exponentially weighted imitation).

## 5 Benchmark Results

Our main empirical finding is that one step of policy improvement is sufficient to beat state of the art results on much of the D4RL benchmark suite [Fu et al., 2020]. This is striking since prior work focuses on iteratively estimating the Q function of the current policy iterate, but we only use one step derived from $\widehat{Q}^\beta$. Results are shown in Table 1. Full experimental details are in Appendix C and code can be found at `https://github.com/davidbrandfonbrener/onestep-rl`.

Table 1: Results of one-step algorithms on the D4RL benchmark. The first column gives the best results across several iterative algorithms considered in Fu et al. [2020]. Each algorithm is tuned over 6 values of their respective hyperparameter. We report the mean and standard error over 10 seeds of the training process and using 100 evaluation episodes per seed. We **bold** the best result on each dataset and blue any result where a one-step algorithm beat the best reported iterative result from Fu et al. [2020]. We use m for medium, m-e for medium-expert, m-re for medium-replay, r for random, and c for cloned.

| | Iterative | One-step | | | |
|---|---|---|---|---|---|
| | Fu et al. [2020] | BC | Easy BCQ | Rev. KL Reg | Exp. Weight |
| halfcheetah-m | 46.3 | $42.1 \pm 0.1$ | $52.6 \pm 0.1$ | $\mathbf{55.6 \pm 0.2}$ | $48.6 \pm 0.0$ |
| walker2d-m | 81.1 | $70.2 \pm 1.3$ | $\mathbf{86.9 \pm 0.4}$ | $85.6 \pm 0.4$ | $80.3 \pm 1.1$ |
| hopper-m | 58.8 | $49.8 \pm 0.6$ | $69.7 \pm 2.1$ | $\mathbf{83.3 \pm 1.4}$ | $56.7 \pm 0.8$ |
| halfcheetah-m-e | 64.7 | $60.1 \pm 0.8$ | $77.0 \pm 0.9$ | $\mathbf{93.5 \pm 0.1}$ | $91.7 \pm 0.9$ |
| walker2d-m-e | 111.0 | $93.6 \pm 5.6$ | $111.8 \pm 0.2$ | $110.9 \pm 0.1$ | $\mathbf{112.9 \pm 0.2}$ |
| hopper-m-e | **111.9** | $48.1 \pm 1.5$ | $81.4 \pm 1.9$ | $102.1 \pm 1.3$ | $83.1 \pm 7.0$ |
| halfcheetah-m-re | **47.7** | $34.9 \pm 0.3$ | $38.4 \pm 0.3$ | $42.4 \pm 0.1$ | $38.6 \pm 0.5$ |
| walker2d-m-re | 26.7 | $23.9 \pm 1.6$ | $66.4 \pm 2.0$ | $\mathbf{71.6 \pm 3.1}$ | $49.3 \pm 3.5$ |
| hopper-m-re | 48.6 | $21.2 \pm 1.3$ | $77.3 \pm 2.7$ | $71.0 \pm 8.1$ | $\mathbf{94.1 \pm 2.4}$ |
| halfcheetah-r | **35.4** | $2.2 \pm 0.0$ | $5.4 \pm 0.1$ | $6.9 \pm 1.0$ | $3.7 \pm 0.2$ |
| walker2d-r | **7.3** | $0.7 \pm 0.1$ | $4.2 \pm 0.2$ | $6.1 \pm 0.3$ | $5.2 \pm 0.2$ |
| hopper-r | **12.2** | $2.6 \pm 0.4$ | $6.7 \pm 0.1$ | $7.8 \pm 0.3$ | $5.6 \pm 0.6$ |
| pen-c | 56.9 | $49.3 \pm 2.2$ | $\mathbf{67.0 \pm 1.1}$ | $55.3 \pm 1.9$ | $54.7 \pm 2.3$ |
| hammer-c | 2.1 | $0.5 \pm 0.1$ | $\mathbf{2.8 \pm 0.5}$ | $0.2 \pm 0.0$ | $1.2 \pm 0.2$ |
| relocate-c | -0.1 | $0.0 \pm 0.0$ | $\mathbf{0.3 \pm 0.0}$ | $0.1 \pm 0.0$ | $0.1 \pm 0.0$ |
| door-c | **0.4** | $0.0 \pm 0.0$ | $0.4 \pm 0.2$ | $0.0 \pm 0.1$ | $0.1 \pm 0.1$ |

As we can see in the table, all of the one-step algorithms usually outperform the best iterative algorithms tested by Fu et al. [2020]. The one notable exception is the case of random data (especially on halfcheetah), where iterative algorithms have a clear advantage. We will discuss potential causes of this further in Section 7.

To give a more direct comparison that controls for any potential implementation details, we use our implementation of reverse KL regularization to create multi-step and iterative algorithms. We are not using algorithmic modifications like Q ensembles, regularized Q values, or early stopping that have been used in prior work. But, our iterative algorithm recovers similar performance to prior regularized actor-critic approaches. These results are shown in Table 2.

Table 2: Results of reverse KL regularization on the D4RL benchmark across one-step, multi-step, and iterative algorithms. Again we run 6 hyperparameters and report the mean and standard error across 10 seeds using 100 evaluation episodes.

|  | One-step | Multi-step | Iterative |
|---|---|---|---|
| halfcheetah-m | **55.6 ± 0.2** | 40.8 ± 8.6 | 47.4 ± 3.5 |
| walker2d-m | **85.6 ± 0.4** | 75.9 ± 0.5 | 75.4 ± 0.8 |
| hopper-m | **83.3 ± 1.4** | 53.0 ± 1.0 | 54.2 ± 0.6 |
| halfcheetah-m-e | **93.5 ± 0.1** | **93.6 ± 0.3** | **93.6 ± 0.2** |
| walker2d-m-e | **110.9 ± 0.1** | 76.3 ± 15.9 | 108.2 ± 0.3 |
| hopper-m-e | **102.1 ± 1.3** | **101.3 ± 3.9** | 82.7 ± 7.4 |
| halfcheetah-r | 6.9 ± 1.0 | 13.7 ± 1.7 | **16.3 ± 1.6** |
| walker2d-r | **6.1 ± 0.3** | 5.0 ± 0.3 | 5.1 ± 0.3 |
| hopper-r | 7.8 ± 0.3 | **15.4 ± 2.9** | 9.7 ± 0.1 |

Put together, these results immediately suggest some guidance to the practitioner: it is worthwhile to run the one-step algorithm as a baseline before trying something more elaborate. The one-step algorithm is substantially simpler than prior work, but frequently achieves better performance.

## 6   What goes wrong for iterative algorithms?

The benchmark experiments show that one step of policy improvement often beats iterative and multi-step algorithms. In this section we dive deeper to understand why this happens. First, by examining the learning curves of each of the algorithms we note that iterative algorithms require stronger regularization to avoid instability. Then we identify two causes of this instability: *distribution shift* and *iterative error exploitation*.

Distribution shift causes evaluation error by reducing the effective sample size in the fixed dataset for evaluating the current policy and has been extensively considered in prior work as discussed below. Iterative error exploitation occurs when we repeatedly optimize policies against our Q estimates and exploit their errors. This introduces a bias towards overestimation at each step (much like the training error in supervised learning is biased to be lower than the test error). Moreover, by iteratively re-using the data and using prior Q estimates to warmstart training at each step, the errors from one step are amplified at the next. This type of error is particular to multi-step and iterative algorithms.

### 6.1   Learning curves and hyperparameter sensitivity

To begin to understand why iterative and multi-step algorithms can fail it is instructive to look at the learning curves. As shown in Figure 2, we often observe that the iterative algorithm will begin to learn and then crash. Regularization can help to prevent this crash since strong enough regularization towards the behavior policy ensures that the evaluation is nearly on-policy.

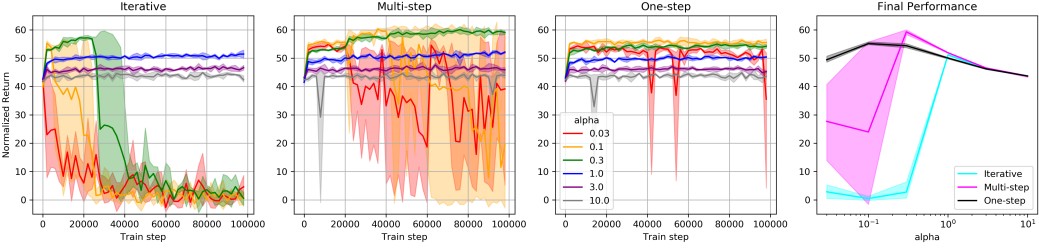

Figure 2: Learning curves and final performance on halfcheetah-medium across different algorithms and regularization hyperparameters (all using the reverse KL regularized improvement operator). Error bars show min and max over 3 seeds. Similar figures for other datasets from D4RL can be found in Appendix D.

In contrast, the one-step algorithm is more robust to the regularization hyperparameter. The rightmost panel of the figure shows this clearly. While iterative and multi-step algorithms can have their performance degrade very rapidly with the wrong setting of the hyperparameter, the one-step approach is more stable. Moreover, we usually find that the optimal setting of the regularization hyperparameter is lower for the one-step algorithm than the iterative or multi-step approaches.

## 6.2 Distribution shift

Any algorithm that relies on off-policy evaluation will struggle with distribution shift in the evaluation step. Trying to evaluate a policy that is substantially different from the behavior reduces the effective sample size and increases the variance of the estimates. Explicitly, by distribution shift we mean the shift between the behavior distribution (the distribution over state-action pairs in the dataset) and the evaluation distribution (the distribution that would be induced by the policy $\pi$ we want to evaluate).

**Prior work.** There is a substantial body of prior theoretical work that suggests that off-policy evaluation can be difficult and this difficulty scales with some measure of distribution shift. Wang et al. [2020a], Amortila et al. [2020], Zanette [2021] give exponential (in horizon) lower bounds on sample complexity in the linear setting even with good feature representations that can represent the desired Q function and assuming good data coverage. Upper bounds generally require very strong assumptions on both the representation and limits on the distribution shift [Wang et al., 2021, Duan et al., 2020, Chen and Jiang, 2019]. Moreover, the assumed bounds on distribution shift can be exponential in horizon in the worst case. On the empirical side, Wang et al. [2021] demonstrates issues with distribution shift when learning from pre-trained features and provides a nice discussion of why distribution shift causes error amplification. Fujimoto et al. [2018a] raises a similar issue under the name "extrapolation error". Regularization and constraints are meant to reduce issues stemming from distribution shift, but also reduce the potential for improvement over the behavior.

**Empirical evidence.** Both the multi-step and iterative algorithms in our experiments rely on off-policy evaluation as a key subroutine. We examine how easy it is to evaluate the policies encountered along the learning trajectory. To control for issues of iterative error exploitation (discussed in the next subsection), we train Q estimators from scratch on a heldout evaluation dataset sampled from the behavior policy. We then evaluate these trained Q function on rollouts from 1000 datapoints sampled from the replay buffer. Results are shown in Figure 3.

The results show a correlation betweed KL and MSE. Moreover, we see that the MSE generally increases over training. One way to mitigate this, as seen in the figure, is to use a large value of $\alpha$. We just cannot take a very large step before running into problems with distribution shift. But, when we take such a small step, the information from the on-policy $\widehat{Q}^{\beta}$ is about as useful as the newly estimated $\widehat{Q}^{\pi}$. This is seen, for example, in Figure 2 where we get very similar performance across algorithms at high levels of regularization.

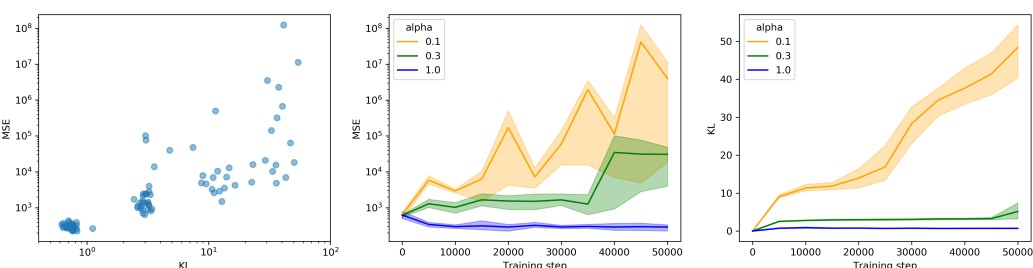

Figure 3: Results of running the iterative algorithm on halfcheetah-medium. Each checkpointed policy is evaluated by a Q function trained from scratch on heldout data. MSE refers to $\mathbb{E}_{s,a\sim\beta}[(\hat{Q}^{\pi_i}(s,a) - Q^{\pi_i}(s,a))^2]$ and KL refers to $\mathbb{E}_{s\sim\beta}[KL(\pi(\cdot|s)\|\beta(\cdot|s)]$. Left: 90 policies taken from various points in training with various hyperaparmeters and random seeds. Center: MSE learning curves. Right: KL learning curves. Error bars show min and max over 3 random seeds.

### 6.3 Iterative error exploitation

The previous subsection identifies how any algorithm that uses off-policy evaluation is fundamentally limited by distribution shift, even if we were given fresh data and trained Q functions from scratch at every iteration. But, in practice, iterative algorithms repeatedly iterate between optimizing policies against estimated Q functions and re-estimating the Q functions using the *same data* and using the Q function from the previous step to warm-start the re-estimation. This induces dependence between steps that causes a problem that we call iterative error exploitation.

**Intuition about the problem.** In short, iterative error exploitation happens because $\pi_i$ tends to choose overestimated actions in the policy improvement step, and then this overestimation propagates via dynamic programming in the policy evaluation step. To illustrate this issue more formally, consider the following: at each $s, a$ we suffer some Bellman error $\varepsilon_\beta^\pi(s, a)$ based on our fixed dataset collected by $\beta$. Formally,

$$\widehat{Q}^\pi(s, a) = r(s, a) + \gamma \mathop{\mathbb{E}}_{\substack{s'|s,a \\ a'\sim\pi|s'}} [\widehat{Q}^\pi(s', a')] + \varepsilon_\beta^\pi(s, a). \tag{5}$$

Intuitively, $\varepsilon_\beta^\pi$ will be larger at state-actions with less coverage in the dataset collected by $\beta$. Note that $\varepsilon_\beta^\pi$ can absorb all error whether it is caused by the finite sample size or function approximation error.

All that is needed to cause iterative error exploitation is that the $\epsilon_\beta^\pi$ are highly correlated across different $\pi$, but for simplicity, we will assume that $\varepsilon_\beta^\pi$ is *the same* for all policies $\pi$ estimated from our fixed offline dataset and instead write $\varepsilon_\beta$. Now that the errors do not depend on the policy we can treat the errors as auxiliary rewards that obscure the true rewards and see that

$$\widehat{Q}^\pi(s, a) = Q^\pi(s, a) + \widetilde{Q}_\beta^\pi(s, a), \qquad \widetilde{Q}_\beta^\pi(s, a) := \mathop{\mathbb{E}}_{\pi|s_0,a_0=s,a} \left[ \sum_{t=0}^\infty \gamma^t \varepsilon_\beta(s_t, a_t) \right]. \tag{6}$$

This assumption is somewhat reasonable since we expect the error to primarily depend on the data. And, when the prior Q function is used to warm-start the current one (as is generally the case in practice), the approximation errors are automatically passed between steps.

Now we can explain the problem. Recall that under our assumption the $\varepsilon_\beta$ are fixed once we have a dataset and likely to have larger magnitude the further we go from the support of the dataset. So, with each step $\pi_i$ is able to better maximize $\varepsilon_\beta$, thus moving further from $\beta$ and increasing the magnitude of $\widetilde{Q}_\beta^{\pi_i}$ relative to $Q^{\pi_i}$. Even though $Q^{\pi_i}$ may provide better signal than $Q^\beta$, it can easily be drowned out by $\widetilde{Q}_\beta^{\pi_i}$. In contrast, $\widetilde{Q}_\beta^\beta$ has small magnitude, so the one-step algorithm is robust to errors[1].

**An example.** Now we consider a simple gridworld example to illustrate iterative error exploitation. This example fits exactly into the setup outlined above since all errors are due to reward estimation so the $\varepsilon_\beta$ is indeed constant over all $\pi$. The gridworld we consider has one deterministic good state with reward 1 and many stochastic bad states that have rewards distributed as $\mathcal{N}(-0.5, 1)$. We collect a dataset of 100 trajectories, each of length 100. One run of the multi-step offline regularized policy iteration algorithm is illustrated in Figure 4.

In the example we see that one step often outperforms multiple steps of improvement. Intuitively, when there are so many noisy states, it is likely that a few of them will be overestimated. Since the data is re-used for each step, these overestimations persist and propagate across the state space due to iterative error exploitation. This property of having many bad, but poorly estimated states likely also exists in the high-dimensional control problems encountered in the benchmark where there are many ways for the robots to fall down that are not observed in the data for non-random behavior. Moreover, both settings have larger errors in areas where we have less data. So even though the errors in the gridworld are caused by noise in the rewards, while errors in D4RL are caused by function approximation, we think this is a useful mental model of the problem.

---

[1]We should note that iterative error exploitation is similar to the overestimation addressed by double Q learning [Van Hasselt et al., 2016, Fujimoto et al., 2018b], but distinct. Since we are in the offline setting, the errors due to our finite dataset can be iteratively exploited more and more, while in the online setting considered by double Q learning, fresh data prevents this issue. We are also considering an algorithm based on policy iteration rather than value iteration.

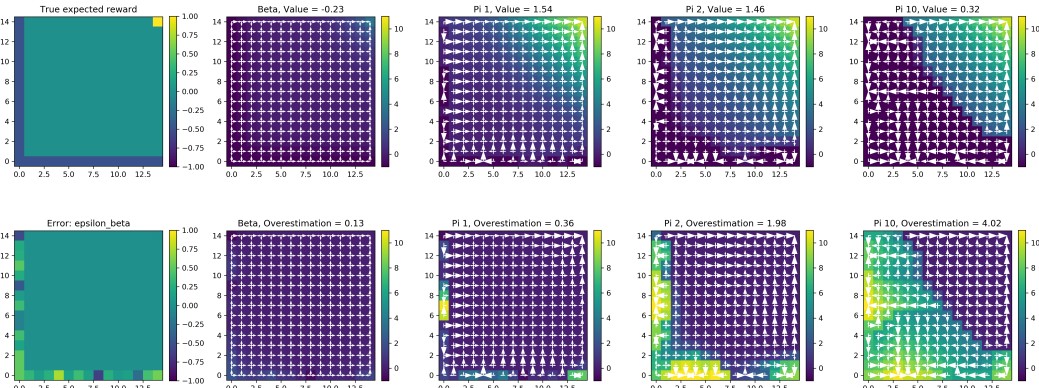

Figure 4: An illustration of multi-step offline regularized policy iteration. The leftmost panel in each row shows the true reward (top) or error $\varepsilon_\beta$ (bottom). Then each subsequent panel plots $\pi_i$ (with arrow size proportional to $\pi_i(a|s)$) over either $Q^{\pi_i}$ (top) or $\widetilde{Q}_\beta^\pi$ (bottom), averaged over actions at each state. The one-step policy ($\pi_1$) has the highest value. The behavior policy here is a mixture of optimal $\pi^*$ and uniform $u$ with coefficient 0.2 so that $\beta = 0.2 \cdot \pi^* + 0.8 \cdot u$. We set $\alpha = 0.1$ as the regularization parameter for reverse KL regularization.

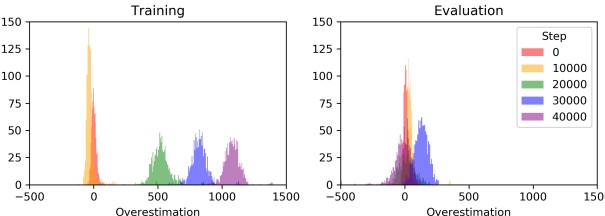

Figure 5: Histograms of overestimation error $(\widehat{Q}^{\pi_i}(s,a) - Q^{\pi_i}(s,a))$ on halfcheetah-medium with the iterative algorithm. Left: errors from the training Q function. Right: errors from an independently trained Q function.

**Empirical evidence.** In practice we cannot easily visualize the progression of errors. However, the dependence between steps still arises as overestimation of the Q values. We can track the overestimation of the Q values over training as a way to measure how much bias is being induced by optimizing against our dependent Q estimators. As a control we can also train Q estimators from scratch on independently sampled evaluation data. These independently trained Q functions do not have the same overestimation bias even though the squared error does tend to increase as the policy moves further from the behavior (as seen in Figure 3). Explicitly, we track 1000 state, action pairs from the replay buffer over training. For each checkpointed policy we perform 3 rollouts at each state to get an estimate of the true Q value and compare this to the estimated Q value. Results are shown in Figure 5.

## 7 When are multiple steps useful?

So far we have focused on why the one-step algorithm often works better than the multi-step and iterative algorithms. However, we do not want to give the impression that one-step is always better. Indeed, our own experiments in Section 5 show a clear advantage for the multi-step and iterative approaches when we have randomly collected data. While we cannot offer a precise delineation of when one-step will outperform multi-step, in this section we offer some intuition as to when we can expect to see benefits from multiple steps of policy improvement.

As seen in Section 6, multi-step and iterative algorithms have problems when they propagate estimation errors. This is especially problematic in noisy and/or high dimensional environments. While the multi-step algorithms propagate this noise more widely than the one-step algorithm, they also

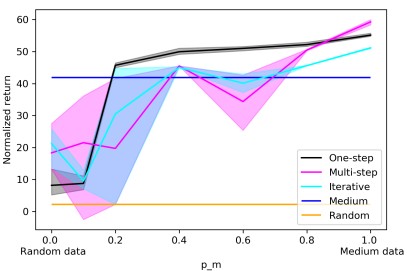

Figure 6: Performance of all three algorithms with reverse KL regularization across mixtures between halfcheetah-random and halfcheetah-medium. Error bars indicate min and max over 3 seeds.

propagate the signal. So, when we have sufficient coverage to reduce the magnitude of the noise, this increased propagation of signal can be beneficial. The D4RL experiments suggest that we are usually on the side of the tradeoff where the errors are large enough to make one-step preferable.

In Appendix A we illustrate a simple gridworld example where a slight modification of the behavior policy from Figure 4 makes multi-step dramatically outperform one-step. This modified behavior policy (1) has better coverage of the noisy states (which reduces error, helping multi-step), and (2) does a worse job propagating the reward from the good state (hurting one-step).

We can also test empirically how the behavior policy effects the tradeoff between error and signal propagation. To do this we construct a simple experiment where we mix data from the random behavior policy with data from the medium behavior policy. Explicitly we construct a dataset $D$ out of the datasets $D_r$ for random and $D_m$ for medium such that each trajectory in $D$ comes from the medium dataset with probability $p_m$. So for $p_m = 0$ we have the random dataset and $p_m = 1$ we have the medium dataset, and in between we have various mixtures. Results are shown in Figure 6. It takes surprisingly little data from the medium policy for one-step to outperform the iterative algorithm.

# 8 Discussion, limitations, and future work

This paper presents the surprising effectiveness of a simple one-step baseline for offline RL. We examine the failure modes of iterative algorithms and the conditions where we might expect them to outperform the simple one-step baseline. This provides guidance to a practitioner that the simple one-step baseline is a good place to start when approaching an offline RL problem.

But, we leave many questions unanswered. One main limitation is that we lack a clear theoretical characterization of which environments and behaviors can guarantee that one-step outperforms multi-step or visa versa. Such results will likely require strong assumptions, but could provide useful insight. We don't expect this to be easy as it requires understanding policy iteration which has been notoriously difficult to analyze, often converging much faster than the theory would suggest [Sutton and Barto, 2018, Agarwal et al., 2019]. Another limitation is that while only using one step is perhaps the simplest way to avoid the problems of off-policy evaluation, there are possibly other more elaborate algorithmic solutions that we did not consider here. However, our strong empirical results suggest that the one-step algorithm is at least a strong baseline.

**Broader impact.** Our paper studies a simple and effective baseline approach to the offline RL problem. The effectiveness of this baseline raises some serious questions about the utility of prior work proposing substantially more complicated methods. By making this observation of prior shortcomings, our paper has the potential to encourage researchers to derive new and better methods for offline RL. This has many potential impacts on fields as diverse as robotics and healthcare where better offline decision making can lead to better real-world performance. As always, we note that machine learning improvements come in the form of "building machines to do X better". For a sufficiently malicious or ill-informed choice of X, almost any progress in machine learning might indirectly lead to a negative outcome, and our work is not excluded from that.

**Acknowledgements**

This work is partially supported by the Alfred P. Sloan Foundation, NSF RI-1816753, NSF CAREER CIF 1845360, NSF CHS-1901091, Samsung Electronics, and the Institute for Advanced Study. DB is supported by the Department of Defense (DoD) through the National Defense Science & Engineering Graduate Fellowship (NDSEG) Program.

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
