# A  Gridworld example where multi-step outperforms one-step

As explained in the main text, this section presents an example that is only a slight modification of the one in Figure 4, but where a multi-step approach is clearly preferred over just one step. The data-generating and learning processes are exactly the same (100 trajectories of length 100, discount 0.9, $\alpha = 0.1$ for reverse KL regularization). The only difference is that rather than using a behavior that is a mixture of optimal and uniform, we use a behavior that is a mixture of maximally suboptimal and uniform. If we call the suboptimal policy $\pi^-$ (which always goes down and left in our gridworld), then the behavior for the modified example is $\beta = 0.2 \cdot \pi^- + 0.8 \cdot u$, where $u$ is uniform. Results are shown in Figure 7.

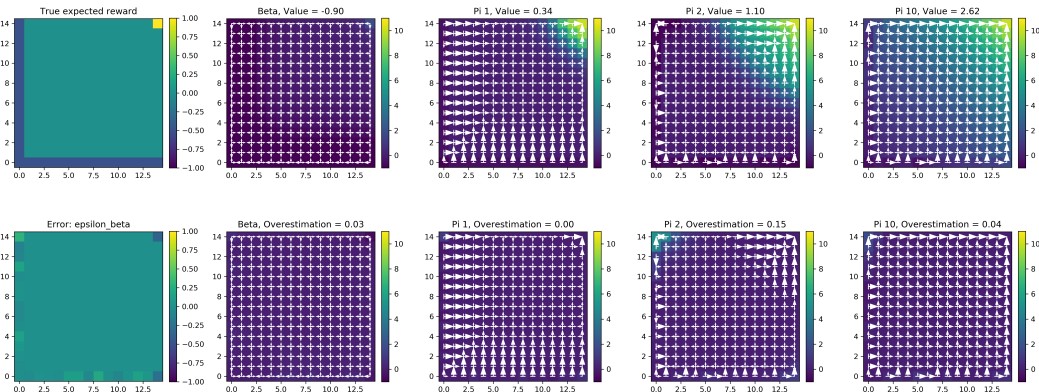

Figure 7: A gridworld example with modified behavior where multi-step is much better than one-step.

By being more likely to go to the noisy states, this behavior policy allows us to get lower variance estimates of the rewards. Essentially, the coverage of the behavior policy in this example reduces the magnitude of the evaluation errors. This allows for more aggressive planning using multi-step methods. Moreover, since the behavior is less likely to go to the good state, the behavior Q function does not propagate the signal from the rewarding state as far, harming the one-step method.

# B  Connection to policy improvement guarantees

The regularized or constrained one-step algorithm performs an update that directly inherits guarantees from the literature on conservative policy improvement [Kakade and Langford, 2002, Schulman et al., 2015, Achiam et al., 2017]. These original papers consider an online setting where more data is collected at each step, but the guarantee at each step applies to our one-step offline algorithm.

The key idea of this line of work begins with the performance difference lemma of Kakade and Langford [2002], and then lower bounds the amount of improvement over the behavior policy. Define the discounted state visitation distribution for a policy $\pi$ by $d^\pi(s) := (1 - \gamma) \sum_{t=0}^\infty \gamma^t \mathbb{P}_{\rho, P, \pi}(s_t = s)$. We will also use the shorthand $Q(s, \pi)$ to denote $\mathbb{E}_{a \sim \pi|s}[Q(s, a)]$. Then we have the performance difference lemma as follows.

**Lemma 1** (Performance difference, Kakade and Langford [2002])**.** *For any two policies $\pi$ and $\beta$,*

$$J(\pi) - J(\beta) = \frac{1}{1 - \gamma} \mathbb{E}_{s \sim d^\pi}[Q^\beta(s, \pi) - Q^\beta(s, \beta)]. \tag{7}$$

Then, Corollary 1 from Achiam et al. [2017] (reproduced below) gives a guarantee for the one-step algorithm. The key idea is that when $\pi$ is sufficiently close to $\beta$, we can use $Q^\beta$ as an approximation to $Q^\pi$.

**Lemma 2** (Conservative Policy Improvement, Achiam et al. [2017])**.** *For any two policies $\pi$ and $\beta$, let $\|A_\pi^\beta\|_\infty = \sup_s |Q^\beta(s, \pi) - Q^\beta(s, \beta)|$. Then,*

$$J(\pi) - J(\beta) \geq \frac{1}{1 - \gamma} \mathbb{E}_{s \sim d^\beta}\left[\left(Q^\beta(s, \pi) - Q^\beta(s, \beta)\right) - \frac{2\gamma \|A_\pi^\beta\|_\infty}{1 - \gamma} D_{TV}(\pi(\cdot|s)\|\beta(\cdot|s))\right] \tag{8}$$

*where $D_{TV}$ denotes the total variation distance.*

Replacing $Q^\beta$ with $\widehat{Q}^\beta$ and the TV distance by the KL (using Pinsker's inequality), we get precisely the objective that we optimize in the one-step algorithm. This shows that the one-step algorithm indeed optimizes a lower bound on the performance difference. Of course, in practice we replace the potentially large multiplier on the divergence term by a hyperparameter, but this theory at least motivates the soundness of the approach.

We are not familiar with similar guarantees for the iterative or multi-step approaches that rely on off-policy evaluation.

## C   Experimental setup

Code for our experimental setup can be found at `https://github.com/davidbrandfonbrener/onestep-rl`.

### C.1   Benchmark experiments (Tables 1 and 2, Figure 2)

**Data.**   We use the datasets from the D4RL benchmark [Fu et al., 2020]. We use the latest versions, which are v2 for the mujoco datasets and v1 for the adroit datasets.

**Hyperparameter tuning.**   We follow the practice of Fu et al. [2020] and tune a small set of hyperparameters by interacting with the simulator to estimate the value of the policies learned under each hyperparameter setting. The hyperparameter sets for each algorithm can be seen in Table 3. We tune hyperparameters using 3 seeds, but then evaluate the best hyperparameter by training on an additional 7 seeds and then report results on the 10 total seeds.

Table 3: Hyperparameter sweeps for each algorithm.

| Algorithm | Hyperparameter set |
|---|---|
| Reverse KL ($\alpha$) | {0.03, 0.1, 0.3, 1.0, 3.0, 10.0} |
| Easy BCQ ($M$) | {2, 5, 10, 20, 50, 100} |
| Exponentially weighted ($\tau$) | {0.1, 0.3, 1.0, 3.0, 10.0, 30.0} |

This may initially seem like "cheating", but can be a reasonable setup if we are considering applications like robotics where we can feasibly test a small number of trained policies on the real system. Also, since prior work has used this setup, it makes it easiest to compare our results if we use it too. While beyond the scope of this work, we do think that better offline model selection procedures will be crucial to make offline RL more broadly applicable. A good primer on this topic can be found in Paine et al. [2020].

**Models.**   All of our Q functions and policies are simple MLPs with ReLU activations and 2 hidden layers of width 1024. Our policies output a truncated normal distribution with diagonal covariance where we can get reparameterized samples by sampling from a uniform distribution and computing the differentiable inverse CDF [Burkhardt, 2014]. We found this to be more stable than the tanh of normal used by e.g. Fu et al. [2020], but to achieve similar performance when both are stable. We use these same models across all experiments.

**One-step training procedure.**   For all of our one-step algorithms, we train our $\hat{\beta}$ behavior estimate by imitation learning for 500k gradient steps using Adam [Kingma and Ba, 2014] with learning rate 1e-4 and batch size 512. We train our $\widehat{Q}^\beta$ estimator by fitted Q evaluation with a target network for 2 million gradient steps using Adam with learning rate 1e-4 and batch size 512. The target is updated softly at every step with parameter $\tau = 0.005$. All policies are trained for 100k steps again with Adam using learning rate 1e-4 and batch size 512.

Easy BCQ does not require training a policy network and just uses $\hat{\beta}$ and $\widehat{Q}^\beta$ to define it's policy. For the exponentially weighted algorithm, we clip the weights at 100 to prevent numerical instability. To estimate reverse KL at some state we use 10 samples from the current policy and the density defined by our estimated $\hat{\beta}$.

Each random seed retrains all three models (behavior, Q, policy) from different initializations. We use three random seeds.

**Multi-step training procedure.**   For multi-step algorithms we use all the same hyperparameters as one-step. We initialize our policy and Q function from the same pre-trained $\hat{\beta}$ and $\widehat{Q}^\beta$ as we use for

the one-step algorithm trained for 500k and 2 million steps respectively. Then we consider 5 policy steps. To ensure that we use the same number of gradient updates on the policy, each step consists of 20k gradient steps on the policy followed by 200k gradient steps on the Q function. Thus, we take the same 100k gradient steps on the policy network. Now the Q updates are off-policy so the next action $a'$ is sampled from the current policy $\pi_i$ rather than from the dataset.

**Iterative training procedure.** For iterative algorithms we again use all the same hyperparameters and initialize from the same $\hat{\beta}$ and $\widehat{Q}^\beta$. We again take the same 100k gradient steps on the policy network. For each step on the policy network we take 2 off-policy gradient steps on the Q network.

**Evaluation procedure.** To evaluate each policy we run 100 trajectories in the environment and compute the mean. We then report the mean and standard error over 10 training seeds.

## C.2 MSE experiment (Figure 3)

**Data.** To get an independently sampled dataset of the same size as the training set, we use the behavior cloned policy $\hat{\beta}$ to sample 1000 trajectories. The checkpointed policies are taken at intervals of 5000 gradient steps from each of the three training seeds.

**Training procedure.** The $\widehat{Q}^{\pi_i}$ training procedure is the same as before so we use Adam with step size 1e-4 and batch size 512 and a target network with soft updates with parameter 0.005. We train for 1 million steps.

**Evaluation procedure.** To evaluate MSE, we sample 1000 state, action pairs from the original training set and from each state, action pair we run 3 rollouts. We take the mean over the rollouts and then compute squared error at each state, action pair and finally get MSE by taking the mean over state, action pairs. The reported reverse KL is evaluated by samples during training. At each state in a batch we take 10 samples to estimate the KL at that state and then take the mean over the batch.

## C.3 Gridworld experiment (Figure 4)

**Environment.** The environment is a 15 x 15 gridworld with deterministic transitions. The rewards are deterministically 1 for all actions taken from the state in the top right corner and stochastic with distribution $\mathcal{N}(-0.5, 1)$ for all actions taken from states on the left or bottom walls. The initial state is uniformly random. The discount is 0.9.

**Data.** We collect data from a behavior policy that is a mixture of the uniform policy (with probability 0.8) and an optimal policy (with probability 0.2). We collect 100 trajectories of length 100.

**Training procedure.** We give the agent access to the deterministic transitions. The only thing for the agent to do is estimate the rewards from the data and then learn in the empirical MDP. We perform tabular Q evaluation by dynamic programming. We initialize with the empirical rewards and do 100 steps of dynamic programming with discount 0.9. Regularized policy updates are solved for exactly by setting $\pi_i(a|s) \propto \beta(a|s) \exp(\frac{1}{\alpha} \widehat{Q}^{\pi_{i-1}}(s, a))$.

## C.4 Overestimation experiment (Figure 5)

This experiment uses the same setup as the MSE experiment. The main difference is we also consider the Q functions learned during training and demonstrate the overestimation relative to the Q functions trained on the evaluation dataset as in the MSE experiment.

## C.5 Mixed data experiment (Figure 6)

We construct datasets with $p_m = \{0.0, 0.1, 0.2, 0.4, 0.6, 0.8, 1.0\}$ by mixing the random and medium datasets from D4RL and then run the same training procedure as we did for the benchmark experiments. Each dataset has the same size, but a different proportion of trajectories from the medium policy.

# D Learning curves

In this section we reproduce the learning curves and hyperparameter plots across the one-step, multi-step, and iterative algorithms with reverse KL regularization, as in Figure 2.

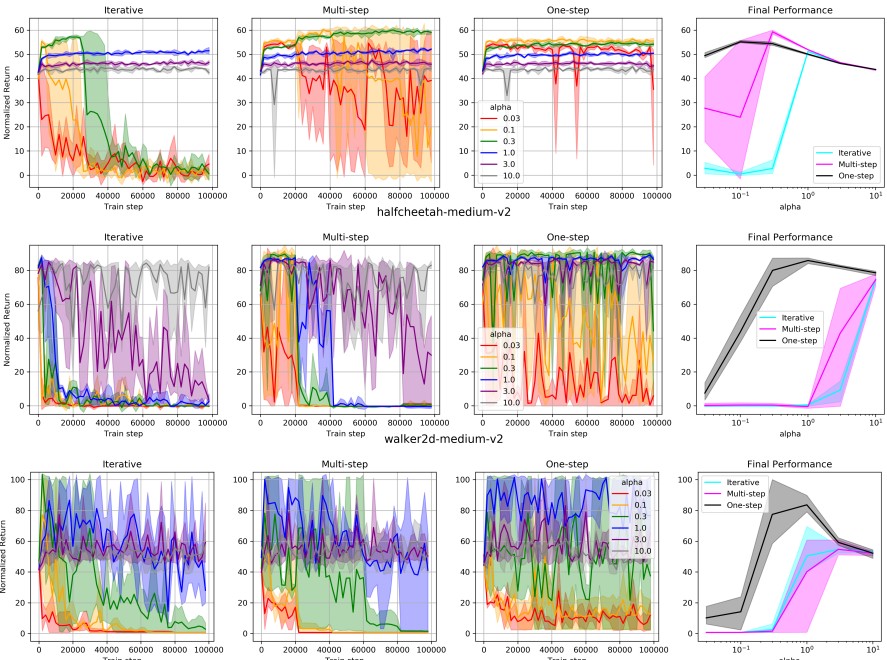

Figure 8: Learning curves on the medium datasets.

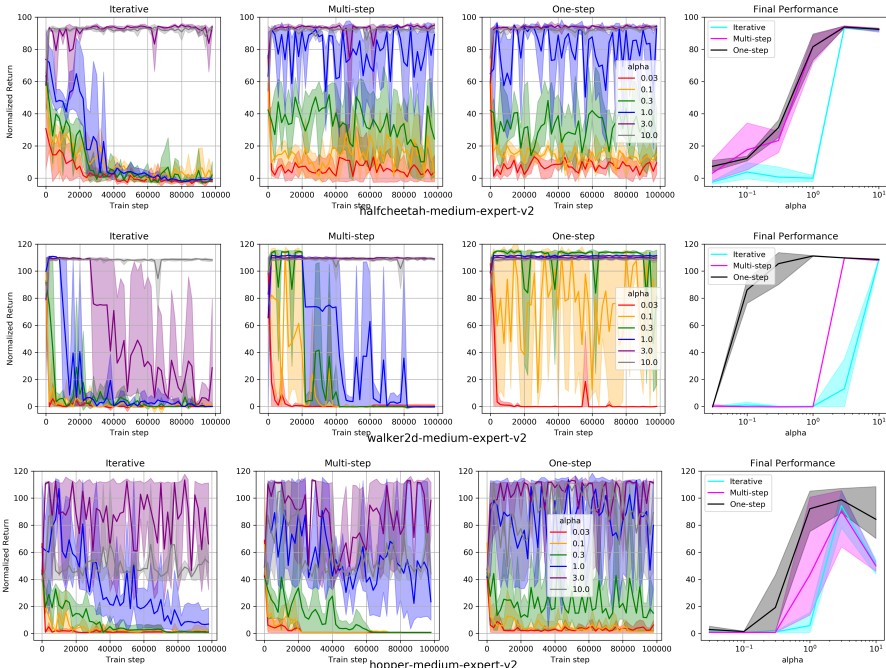

Figure 9: Learning curves on the medium-expert datasets.

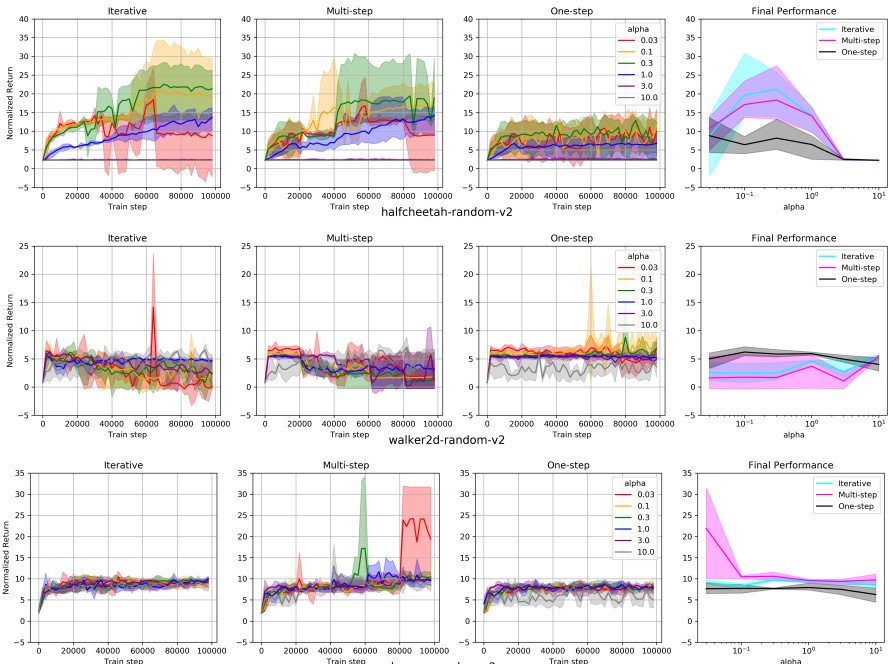

Figure 10: Learning curves on the random datasets.